# Novel MRI Techniques Identifying Vascular Leak and Paravascular Flow Reduction in Early Alzheimer Disease

**DOI:** 10.3390/biomedicines8070228

**Published:** 2020-07-20

**Authors:** Charles R Joseph

**Affiliations:** Department of Internal Medicine, Liberty University College of Osteopathic Medicine, Lynchburg, VA 24502, USA; crjoseph@liberty.edu

**Keywords:** preclinical Alzheimer disease, blood–brain barrier, paravascular outflow, glymphatic system, high definition dynamic contrast enhanced (DCE) imaging MRI, 3D pulsed arterial spin labeling (PASL) MRI

## Abstract

With beta amyloid and tau antibody treatment trial failures, avenues directed to other facets of the disease pathophysiology are being explored to treat in the preclinical or early clinical state. Clear evidence of blood–brain barrier (BBB) breakdown occurring early in the AD process has recently been established. Likewise, the glymphatic system regulating water and solute inflow and outflow in parallel with the vascular system is affected causing delayed clearance of fluid waste. Its dysfunction as a component of AD along with BBB leak are reasonable candidates to explore for future treatments. Ideally, human medication trials require a minimally invasive method of quantifying both improvements in BBB integrity and glymphatic fluid clearance correlated with clinical outcomes. We will review the known physiology and anatomy of the BBB system, and its relationship to the glymphatic system and the microglial surveillance system. Dysfunction of this tripart system occurring in preclinical Alzheimer disease (AD) will be reviewed along with existing MRI tools for identifying altered flow dynamics useful for monitoring improved functionality with future treatments. High-resolution dynamic contrast enhanced MRI imaging demonstrating BBB leak and the recently reported non-invasive 3D PASL MRI pilot study demonstrating significant delay in glymphatic clearance in AD subjects appear to be the best candidates.

## 1. Introduction

Future successful treatment and mitigation of neurodegenerative diseases such as AD will likely need to include treatment of disease processes that may initiate, compound, and accelerate the destructive effects of accumulating misfolded prion-like proteins.

The central nervous system (CNS) is but 5% of total body weight yet consumes 20% of available cardiac output [1]. This high metabolic activity produces an equal amount of metabolic waste products that must be rapidly and efficiently cleared from the neuropil to maintain optimal functioning. Given the metabolic demands, it is not surprising that 85% of the vascular length in the human brain are capillaries, which freely exchange oxygen and CO_2_, provide water and allow for substrate replenishment, and waste removal [2,3,4]. The elimination of internal waste and prevention of outside toxin entry into the neuropil allows for a stable metabolic environment required for favorable neuronal functioning.

Alzheimer disease is the most common cause (60–65%) of dementia. The clinical diagnosis unlike the pathologic diagnosis is based on the pattern and progression of cognitive impairment, ascertained by neuropsychiatric (NP) testing. The cognitive dysfunction classically begins with short term memory impairment and loss of executive function that progress chronically over years. Brain tumor, stroke, inflammatory, infectious, metabolic, vitamin deficiencies, or systemic disorders must be excluded as potential causes of dementia. Currently presence or absence of the seminal pathologic changes of accumulated Aβ or HpTau serologically, in cerebral spinal fluid (CSF), or by imaging is not required. Given the absence of effective treatment, the more rigorous pathologic diagnosis given its greater expense and potential harm to patients is unnecessary. Inclusion in AD treatment trials however requires more exacting patient screening to assure the correct pathologic diagnosis for study entry, with accumulated Hp tau and amyloid β-protein oligomers (Aβ) confirmed by testing [5,6,7,8]. Optimal future AD treatment strategy should be focused on the preclinical or early clinical state, before major irreversible Hp tau-related neurodegeneration develops.

The pathophysiology of AD is a multifactorial process with recent evidence demonstrating an initial breach of the BBB by either external causes (e.g., aging process, vasculopathy, etc.,) in sporadic AD or internal causes (e.g., homozygous APOE 4) in hereditary disease. Either way, set in motion are multiple simultaneous or sequential detrimental processes resulting in reduced availability of metabolic substrate (e.g., reduced glucose transport), influx of neuro toxins and cells from blood (e.g., blood borne recycled amyloid β-protein, free iron, RBCs and inflammatory cells), and reduced clearance of metabolic waste including detrimental proteins [2,9,10,11]. This ultimately leads to development, accumulation, and spread of toxic misfolded proteins (Aβ and Hp tau) and neurodegeneration, all required for pathologic diagnosis. The latter occurrence leads to dysfunction and death of neurons and supporting glia with loss of cognitive function [12,13,14,15]. The paradoxical disconnect of classic AD pathologic changes without clinical dementia, and the converse progressive dementia without pathologic AD changes could be explained by the presence or absence of disruption to normal BBB and glymphatic clearance [16,17,18,19,20]. This article summarizes the current extensive work investigating each of the three facets; controlled parenchymal metabolite access (BBB), waste clearance (glymphatic system), and immune surveillance (microglia), and their interrelationships and contribution to the progression of AD. The challenge is finding treatments to effect repair or arrest the disease in the preclinical or early phases. In order to make early diagnosis and monitor effectiveness of treatment, further development of new non-invasive tools is required. New MRI techniques of identifying preclinical BBB leak and diminished glymphatic flow clearance are presented below with potential clinical research strategies for monitoring effectiveness of modifying treatments in the earliest phases of AD.

## 2. Regulation of Fluid and Metabolite Inflow and Outflow

Three main players account for maintaining the metabolic and immunologic integrity of the brain parenchyma. There is the blood–brain barrier (fabricated by endothelial cells and pericytes), the glymphatic system (composed of the paravascular channels, interstitium, and astrocytes), and immune surveillance by microglia/perivascular macrophages (Figure 1). This triumvirate is developmentally and functionally interdependent and as such, must work as a unit to assure proper neuronal function [21,22,23].

## 3. Blood–Brain Barrier

The first inner layer of the blood–brain barrier is composed of endothelial cells and tight inter-endothelial junctions composed of proteins (i.e., vascular endothelial cadherins and other endothelial transmembrane and junctional adhesion molecules), which exclude intravascular solute, proteins, and cellular transmigration into the interstitium [2,24]. The next layer is composed of a fused dual basement membrane, with the outer layer of astrocyte end feet origin and the inner layer of endothelial origin [25]. The two layers are fused at the capillary level but separate at the takeoff of the venules. They serve as framework for the tight junctions and are damaged by matrix metalloprotease release [22,26].

The third layer is pericytes which manage transport of metabolic substrates via endothelial expression of transporters for glucose solute, electrolytes, lipids, and selected proteins, thereby excluding “danger associated molecular patterns” (DAMPS) [21,27,28,29]. Endothelial expression of tight junction proteins is under the control of pericytes which regulate both formation thereof and are the gatekeepers allowing fluid filled vesicle transcytosis by controlling tight junction opening [27,28]. Reciprocally, pericytes themselves are regulated via endothelial cell release of PDGF-BB ligand which binds to pericyte PDGF Beta receptors, facilitating their proliferation, migration to the vascular endothelial wall, and survival [27,28]. Pericyte vascular coverage extends from the pre-capillary arterioles, enveloping capillaries and terminating at the post-capillary venules [29,30,31,32]. They control capillary flow directly by altering luminal diameter by their internal contractile proteins under the control of astrocyte signaling transduction [30]. Regulation of arteriolar tone is via astrocytes and direct neuronal signaling created by local CNS demand through metabolite-derived neurovascular coupling (metabotropic glutamate receptor activity) [33,34]. The blood–brain barrier itself requires the interplay of both pericytes and endothelial cells under the influence of astrocyte expressed TGF-B and receptor for gap junction Connexin-43 hemichannel formation [22,30].

Angiogenesis, differentiation and survivability are regulated by the co-dependent release of pericyte Notch3 and endothelial Notch1/4 receptors and concomitant release and transduction of VGEF-A from both cell types [28]. Likewise, pericyte expression of Nng1 and transduction through endothelial Tie2 pathways also enhances endothelial sprouting and migration [31].

Interaction of astrocytes and pericytes is exemplified by the loss of BBB from pericyte transduction of astrocyte expression of APOE4 caused by upregulation of matrix metalloproteinse-9 (MMP-9) enzymes thereby allowing for local BBB leak [29,35]. Pericytes control leukocyte transmigration into the interstitium indirectly through control of endothelial expression of leukocyte adhesion molecules which allow access into the neuropil [2,31,36]. Pericytes also express receptors for waste removal (e.g., low-density lipoprotein receptor-related protein 1 (LRP1)), which bind microglia bound soluble amyloid β-protein oligomers as well as receptors for advanced glycation end products (RAGE) which paradoxically allow ingress of AB into the parenchyma [37,38]. Thus, the tight junction composition and expression is dependent upon the signaling interplay of pericytes, endothelial cells, and astrocytes [22,28,34,37]. Regulation of capillary blood flow, metabolic substrate, proteins, and lipids as well as waste removal is directed primarily from astrocytes and indirectly by neuron and microglial signaling [34].

The fourth layer separated by the dual layer basement membrane are the astrocyte end feet which cover 98% of the microvasculature and parenchymal basement membrane [38,39]. As they are intimately involved in neuronal maintenance and metabolism, they signal metabolic needs to the pericytes influencing local capillary vascular tone; dilating to increase oxygen and glucose availability when needed (Figure 1).

## 4. Glymphatic System

The importance of waste management is illustrated by the presence of the recently described redundant arrangement to the venous system, namely, the glymphatic system [40,41,42,43,44]. Intertwined with the vascular tree is the glymphatic system which cleverly uses the aquaporin-4 channels expressed in astrocytic end feet to channel ultra-filtered water through the interstitium, and on egress, utilize existing paravascular spaces to channel CSF extraluminally to true lymphatic vessels present within the meninges and dural sinuses [41,43]. Ultimately this fluid flows into the deep cervical lymph nodes and back into general circulation [41]. As mentioned, the gateway into the neuropil for water are the aquaporin 4 (AQ 4) channels expressed in the astrocytic end feet (luminal side). They allow water to freely move into the astrocyte end feet surrounding the capillaries, then diffuse through the interstitium admixed with water produced as a byproduct of oxidative metabolism and diffuse out into the paravascular channels [41,44]. Their expression in the end feet also serves as a conduit for amyloid β-protein egress. With buildup of cerebral vascular amyloid, however, astrocyte end foot retraction of AQ4 occurs [36,38].

A portion of CSF of subarachnoid origin filters into the paravascular spaces via the fenestrated portions of the pial membranes surrounding the deep penetrating arterioles which form the Virchow-Robin spaces [39,42,43]. They are commonly visible by imaging within the Thalamus and more widespread in various disease states. The CSF within the paravascular channels is therefore comingled CSF fluid of choroid plexus origin and glymphatic fluid that has diffused out of the interstitium [42,43,44].

Although controversial, the propulsion mechanism of paravascular flow (PF) has been shown to be by convection, mainly vascular pulsation as well as a respiratory inspiration component [22,43]. Although the flow velocity of PF has not been described, it likely mirrors that of the venous system in the low cm/sec range. Flow through the interstitium, although somewhat controversial, is most likely via diffusion and thus considerably slower in the nm/minute range [44]. The possibility of perivascular flow streaming retrograde within arterial walls has been suggested but has been shown only in animal models with direct infusion of a tracer into the brain parenchyma as opposed to studies infusing intravascular trace where antegrade paravascular flow is demonstrated [42]. Solute and macromolecules are actively transported out of the interstitium [41,42,43]. Impairment of paravascular outflow severely hampers waste removal. Using MRI technology, measurement of outflow dynamics can theoretically be quantified as a clearance rate allowing for comparisons of normal and disease states [45].

## 5. Microglia

The third member of the triumvirate are the microglia/perivascular macrophages that serve to monitor and eliminate ingress of toxic substances and also remove internally produced toxins such as soluble amyloid β-protein oligomers [46]. These cells are stationary like a junk yard dog on a chain, so toxin entry outside of its realm do not attract them. These long-lived cells serve an individual’s lifetime and replenish by mitosis as opposed to via progenitor cells [46,47,48]. In particular, microglia attach to and remove misfolded proteins amyloid β-protein from the ECF into the paravascular space preferentially (80% in one study) and accumulation is avoided [46,47,48,49]. They provide immunologic surveillance and a very localized inflammatory response to presenting antigens [46,47,48,49]. The cell processes are ramified with receptors that bind antigenic substrate and exocytose them via pericytes into the paravascular fluid. The consequence of prolonged microglial longevity is reduced efficiency of waste removal as the cells become mired down with indigestible intracellular inclusions from excessive interactions [47]. The result is accelerated buildup of toxic amyloid β-protein oligomers within the interstitium [39].

## 6. Loss of BBB Integrity in Early AD

The discovery of preclinical low volume BBB leak caused by initial pericyte damage with consequent loss of integrity of tight endothelial junctions, to date is the earliest morphologic change in AD preceding accumulation of Aβ and Hp tau [2,21,37] (Figure 2). The associated dysfunction in metabolic substrate transport, leak in of blood born toxins, and impaired blood flow regulation may be the initial “hit” in the complex development of AD and possibly other neurodegenerative diseases [21]. Whether this is related to focal ischemia or overexpression of pericyte cell surface proteins such as RAGE (diabetes mellitus) or factors as yet unidentified, its occurrence at the sites of initial AD pathological change suggests more than coincidence and rather likely a major component of the disease [2,35]. Given the complex interrelationships of all three components of fluid, metabolite, and waste management, its development must be closely tied temporally to glymphatic flow dysfunction and diminished microglial waste removal. Less clear is whether BBB and glymphatic dysfunction directly or indirectly triggers development of Amyloid and Tau misfolding [50,51,52,53,54]. That question can be answered either indirectly by restoring the systems to normal functioning and observing the effect on misfolded protein accumulation, or directly by defining and manipulating the pathways leading to their maldevelopment.

## 7. The Evidence Demonstrating BBB Dysfunction in Early AD

Several lines of investigation have demonstrated BBB breakdown in AD. Among them are multiple human studies demonstrating plasma-derived proteins, free iron, and cells with hemosiderin deposits from microbleeds, endothelial, and pericyte damage with loss of tight junctions [55]. Further, blood born macrophages and leukocytes from BBB leak have been identified in human AD brain tissue [55,56,57].

Recent developments in neuroimaging using high-resolution dynamic contrast enhanced (DCE) MRI imaging in human subjects with MCI and AD have shown contrast leak into the hippocampus [58]. In mice, pericytes appear to be initially affected confirmed by the presence of pericyte marker PDGF specific tissue marker but not endothelial cell markers (intercellular adhesion molecule-1) or by marker of tight junction loss (matrix metalloproteinase -9) [58]. Only after the pericyte damage was established did they appear and with- it other findings of breached vascular integrity [31]. These changes progressed over time in conjunction with MCI as will be shown by the human MRI studies discussed below. Presence of microbleeds in roughly ½ to ¾ of subjects with MCI or AD demonstrated by susceptibility MRI imaging has been observed in multiple studies, although obfuscating co-morbid vascular disease or prior injury cannot be underestimated [59,60,61,62,63]. The leak of intravascular iron and subsequent triggering of ROS further damages the BBB integrity [64,65].

In association with the loss of BBB integrity is reduced glucose uptake in affected brain regions by (FDG-PET human subjects) due to loss of endothelial expression of the (GLUT1) glucose transporter integral for astrocyte and neuronal function causing functional decline and atrophy [66]. The transgenic mouse model recapitulates these findings [66].

The APP transgenic mouse model of AD has repeated the vascular leak phenomena with similar markers spilling into the parenchyma, with initial pericyte injury, followed by endothelial cells and tight junctions and parenchymal extravasation of blood products [67,68,69,70,71]. These changes precede beta- amyloid and HP Tau deposition [71,72,73]. The subsequent adhesion of circulating leukocytes and endothelial cells enhances the deposition of toxic amyloid oligomers and may ramp up production of both amyloid β-protein and HP Tau [56,57]. That said, any specific trigger linking BBB leak and intraparenchymal synthesis of misfolded amyloid β-protein and hyperphosphorylated Tau has not yet been reported.

With loss of a primary amyloid receptor LRP1 on “sick” pericytes through loss of regulator protein phosphatidylinositol- binding clathrin assembly (PICALM) via human genetic mutation, or efflux transporter Pgp (mice), the clearance of Aβ oligomers is greatly reduced, leading to additional interstitial accumulation [74]. With the loss of barrier integrity, reflux of blood borne amyloid fragments into the interstitium occurs. As a result, inward trafficking by increased expression of RAGE (receptor for advanced glycation end products expressed in diabetes) is observed in injured mice endothelium [75,76].

With this background of timing and effect of loss of BBB integrity in the development of AD, we will consider potential causations categorizing them as extrinsic or intrinsic injury. Since late onset sporadic AD is by far most common, causation of BBB leak can be looked at as a co-morbidity of other disease states most notably microangiopathy (small vessel disease), diabetes with associated expression of RAGE receptors and angiopathy, hypertensive angiopathy, and deleterious advanced age effects on endothelial and pericyte function [2,22,58]. CTE chronic traumatic encephalopathy is yet another pathway for AD like pathology and dementia. All are associated with BBB injury.

Hereditary causes account for about 5% of AD cases and present at an earlier age. Causation is from the inside out, as the most notable causes are related to either APOE 4 protein expression or the presence of presenilin 1 protein [77]. The latter increases production of toxic beta amyloid fragments whereas the former reduces amyloid clearance due to reduced affinity of APOE4 to the LRP1 receptor. In both cases there is ultimate damage to the BBB and leak [33,37,55,78].

The sum total of above makes it clear that loss of BBB integrity is a key component of the AD process. We will move to the second constituent of the disordered neuropil maintenance apparatus, the glymphatic system. Because of the strong interdependencies and signaling of the BBB structures and the glymphatic system, both are likely involved either simultaneously or in rapid succession. With leak of toxic substances in any impairment of toxin egress only amplifies the BBB and parenchymal damage. Our initial investigation suggests paravascular flow is reduced in mild AD [45].

The final component of the clearance system are the microglia that are distressed in late onset AD. In particular, microglia attach to and remove misfolded proteins amyloid β-protein from the interstitium moving them preferentially into the paravascular space (80% in one study) via pericyte LPR-1 receptors, thus accumulation is avoided [46,47,48,49,65].

Given the detrimental effects of both cell longevity and antigen overload by amyloid β-protein oligomers, microglial efficiency is reduced [64].

In summary, although the brain has three methods of waste removal, the consequence of BBB leak beginning with pericyte damage followed by endothelial cell dysfunction, loss of tight junctions, ingress of neurotoxins, loss of capillary flow regulation, and damaged glymphatic egress pathways impairs both proper inflow of nutrients and outflow of waste. These changes both initiate and accelerate the degenerative process. Unexplained is/are the trigger mechanism(s) for the later elaboration of prion-like misfolded proteins (e.g., hyperphosphorylated Tau) in the disease process, the major hallmark of AD [13]. Could, for instance, the barrier leak serve as an entry portal for blood borne prion-like templates of gut or other origin, or heretofore undiscovered toxins activating aberrant post-translational protein modifications? Using the plumbing analogy, a search for BBB “stop leak” and glymphatic “drain cleaner” opens a new line of potential treatment investigations.

To move in this direction, reliable tools for monitoring improvement of both the BBB integrity and outflow system coinciding with clinical outcome measures are required.

## 8. Investigational Imaging Modalities for Preclinical AD Identification

In order to conduct clinical trials in AD, ascertaining the specific pathologic diagnosis for study entrants is critical. The NIA-AA research framework defining the pathologic criterion for AD diagnosis re: the “A-T-N” criterion will serve as a frame of reference [Table 1] [8]. The criterion includes the main pathologic changes of AD, “A” stands for Aβ accumulation, “T” for Hp tau, and “N” for neurodegeneration along with their current respective testing methods.

New MRI tools evaluating the vascular leak or impaired fluid out flow would fit into the “N” category of the A-T-N rubric. That said, for maximal effectiveness, new clinical trials treating very early AD will have to be initiated with less than a complete positive combination of markers but nonetheless suggesting high likelihood of disease development. Since the BBB leak and impaired glymphatic flow may be common to several neurodegenerative syndromes, new confirmatory diagnostic tests for them are required for specific pathologic classification in developing treatment trials. The next issue in preclinical evaluation and treatment is deciding who should be screened and treated. This will largely fall to the epidemiology data of high-risk demographics, such as advanced age, presence of diabetes mellitus or hypertension, previous head trauma etc.

The detection of blood–brain barrier leak and impaired glymphatic clearance requires specific new imaging techniques, perhaps in conjunction with serologic markers of early BBB injury when available. These markers could be coupled with established AD biomarkers, all obtained serially over time.

We will give examples of the three approaches as well as their strengths and weaknesses. For the techniques presented, high field strength MRI (3T or greater) is necessary to obtain satisfactory signal-to-noise ratio (SNR).

## 9. High-Resolution Dynamic Contrast Enhanced MRI

Since minor BBB leak is the sentinel event in AD, we will discuss the MRI technology developed to demonstrate its occurrence first. The challenge is measuring minor increases in signal from low concentrations of residual leaked contrast within (small sub-regions of the hippocampus [58,78,79]. Since the early BBB leak in AD is miniscule in volume compared to the leak incurred from stroke or even multiple sclerosis, quantifying differences requires post processing analysis to be able to discern this minor extravasation of blood components from the intravascular to interstitial spaces. This can be quantitated as a transfer constant (K_transfer_) and assumes insignificant reflux back into circulation. The anatomic site chosen as region of interest also requires understanding the confounding structures contained within that may affect the signal output. For instance, the hippocampal area has choroid plexus which must be avoided when selecting region of interest as increased signal related to its contrast enhancement will exceed the tissues of interest erroneously affecting K_transfer_ [79,80]. Contrast agents themselves add unknowns. For example, the percentage of contrast agent in circulation bound to albumin varies individually and may confound the true transfer rate particularly with minor BBB leaks [78,79,80].

High-resolution DCE-MRI (dynamic contrast enhanced MRI) developed by Barnes et al. utilizes precise modeling of low volume blood to brain transfer rate (K trans) of contrast leaking into the interstitium from the vascular space [76,79]. This pre and post contrast gadolinium sequence allows for parsing small hippocampal subunits demonstrating minute BBB leaks as found in patients with MCI or AD [76,79]. Post-scan analysis uses the Toft’s model for calculating low volume transfer rate [81,82]. Since only small studies have been reported, more experience and a larger patient pool will further validate the technique. Cost and risk of Gadolinium contrast infusion must also be considered. Nonetheless, the high-resolution DCE technique is the most sensitive technique yet reported for identifying and quantitating early BBB leak and deserves further development [76,79,80].

## 10. Arterial Spin Labeling (ASL) Technique

Since the glymphatic system has also been shown to be affected in AD in multiple animal studies and in human perfusion and recent clearance study, this line of MRI investigation will also be discussed. Contrast enhanced and Bold studies have laid the foundation for developing ASL as another means of quantifying both cerebral perfusion and investigating changes in fluid outflow [45,83,84,85]. Arterial spin labeling measurements have several challenges but significant advantages over conventional contrast-enhanced studies [84]. The technology magnetically labels blood in the neck with a timed pulse and after a specified delay time to acquisition, measurements of residual signal in a region of interest (ROI) are obtained. The advantage of using intrinsic blood as a marker eliminates uncertainty related to contrast transfer rates and potential risk of infusion. The amount of residual signal recorded is dependent on the T1 of the tissue where the labeled protons are located at the time of measurement. If a proton remains in blood the T1 (1650ms 3T) is much shorter than if it migrates into free fluid (3800ms 3T) [81,85]. Because the T1 of white matter is 1084ms at 3T and gray matter T1 1820ms at 3T are significantly shorter than T1 of free fluid, minimal residual signal from either is present at long delay to acquisition times. The second issue is knowing blood transit time through the brain (gray matter (MTT) is 2.94 ± 0.52 s, white matter is 3.73 ± 0.60 s), and correlating that with T1 decay times of the compartments of interest. Determination of transit times has been reported using Bold techniques [82]. ASL MRI’s major challenge is low SNR with resultant poor-quality images, and thus quantitating signal requires a large (ROI) to compare from one delay time to the next, to reduce sampling error [45].

## 11. ASL Perfusion

Quantitating perfusion as opposed to clearance is one approach that takes advantage of adequate signal as a single determination [76,86,87,88]. Reduced perfusion theoretically correlates with loss of pericyte regulation of capillary tone [1,28,33]. The advantage is that the T1 of blood is fairly matched with blood flow ingress time, hence more robust SNR is available [78,80]. ROI analysis can be employed to investigate regional differences and compare subjects with disease to normal [76]. The flip side is the shorter interval from proton labeling to inversion acquisition (TI) allows more confounding signal inclusion from gray and white matter. Another confounding variable causing reduced perfusion is presence of co-morbid vasculopathies thus diminishing specificity such as proximal arterial stenosis etc. Correlating well though with reduced regional blood flow, is the reduced glucose utilization found with fluorodeoxyglucose (FDG)-PET imaging [89,90]. The latter speaks to both flow-related reduction of substrate and impaired endothelial expression of glut-1 transporter reducing glucose availability to the neuropil. Whether this is an early or later phenomena in disease progression is unclear.

## 12. 3—D PASL MRI Glymphatic Clearance

Our approach was to consider measuring clearance of labeled protons from regions of interest within the frontal, temporal, and parietal lobes in normal and disease states [46]. This required signal acquisition at longer post- labeling times than used for evaluating perfusion with consequent lower SNR depending on the tissue of interest. The longer delay times reduces confounding signal from white and gray matter given their short T1 values [81]. Our technique investigates the combined clearance rate of fluid-labeled protons in blood (T1 1650 s at 3T) flowing out of the vasculature and labeled protons which remain as CSF (T1 3260 s at 3T) in the interstitium and paravascular space. The combined normal clearance rate is reduced by either leak of water protons into the interstitial space and/or impaired outflow from the paravascular space. If the ratio of labeled protons from CSF increases compared with blood, higher signal will remain in the region of interest (ROI) given the much longer T1_CSF._ By avoiding signal contamination from subarachnoid and ventricular spaces in the ROI, the source of excess labeled CSF protons is labeled fluid leaking into the interstitium (BBB leak) and sequestered there due to impaired paravascular outflow [45].

Among our goals in technique development was ability to use existing approved commercially available sequences. Our community-based 3T magnet was programmed with a 3D PASL sequence, which fulfilled our other objectives of short acquisition times and ease of data storage, and transfer for post study analysis. Certainly stock sequences with pCASL may provide even more signal which may be available on some machines. A brief rundown of the sequence parameters is noted. For a full description see Joseph et al. [45]. Seven serial sequences were obtained at 200 ms inversion time increments. FOV 250 mm × 250 mm, TE 16.36 ms (all sequences), TR 3830 ms with inversion time (TI) 2800 ms; TR 4330 ms with TI 3000 ms; TR 5000 ms with TI 3200 ms; TR 5320 ms for the following TIs 3400 ms, 3600 ms, 3800 ms, and 4000 ms. Please note the TR was increased to accommodate the longer inversion times. QTIPS and background suppression were used. Bolus labeling was 700 ms. Interleaved 40- 4 mm image slices were obtained per sequence with a voxel size of 3.9 × 3.9 × 4 mm^3^. A 64 × 64 × 40 matrix was employed. A 20 channel send receive head coil was used. DICOM (Digital Imaging and Communications in Medicine) per vendor (Seimens) was employed for image reconstruction and transferred to McKesson PACS (picture archiving and communication system) for ROI analysis. The elliptical, manually adjustable tool provided signal average, signal range, and volume studied. The total scan time for the seven sequences was about 20 min, with approximate average scan time per sequence of 2 min 15 s [45].

Choice of labeling to delay times (TI) allowed for linear regression analysis for determining clearance rate (slope of the time dependent signal decay), as the linear regression of T1 decay for both CSF and blood had a 99% correlation with the actual decay curve at the acquisition times chosen [45]. The pitfalls of this technique include low SNR, and avoidance of artifact from ventricular/subarachnoid spaces in the ROI’s chosen over all the acquisition times [45]. Since proton volume during the blood labeling varies among individuals because of pulse rate and volume, we corrected this variation by dividing the signal strength determined at each acquisition time by the patients pulse (taken just prior to the study) [46,91]. Further, since rate of clearance is the slope of the linear decay rate, absolute differences in signal strength intersubject is moot. The corrected signal strength was then plotted against the acquisition times, using linear regression analysis with the slope defining the clearance rate (signal strength (au)/pulse- sec) [47]. Data analysis at this time is not automated but could easily be developed. Importantly, there is 42.7% signal decay from the first to the last acquisition time for T1_blood_ compared to only 22.5% for T1_CSF_ [Appendix A [45]. The bilateral frontal, temporal, and parietal lobes were examined, and all showed delayed clearance rates compared with normal [46]. In this initial study, we were able to find statistically significant reduction in clearance rates in AD subjects compared to the normal in all but one of six brain regions (*p* < 0.038, several *p* < 0.001) (Figure A3 and Figure A4, Table A1) [45]. This means labeled protons (AD subjects) were sequestered as increased free water within paravascular and interstitial spaces [45]. A larger confirming study is underway.

The origin of the sequestered labeled fluid was from BBB leak since inflow through aquaporin channels is lost with their retraction from the astrocyte foot processes, and subarachnoid/ intraventricular spaces were not averaged in the ROI’s. In animal studies of AD, there is withdrawal of AQ4 channels from the luminal astrocytic end feet into the cell body and interstitial abluminal surfaces. This loss of “polarization” reduces ingress and egress of fluid from the paravascular spaces [23]. Thus, the source of resulting accumulation of labeled CSF (AD subjects) in our study points more to BBB leak than Aquaporin 4 source contrary to my original thought [45].

Further development and validation of 3D ASL technique and consideration of combining it with the DCE technique in an image protocol would provide valuable insight into the temporal sequence of developing BBB leak and glymphatic flow dysfunction. If there is high correlation between the two techniques in the early phases of AD, one or the other or both may serve as excellent biomarkers for future treatment research and general clinical use.

## 13. Combined Methodology for Identification of Preclinical AD

It cannot be emphasized enough that in order to effectively treat AD, treatment must begin early to prevent Hp tau-related neurodegeneration. The underpinnings of the disease process must be addressed, specifically the BBB leak and impaired glymphatic clearance.

Organizing a coherent testing strategy in evaluation of preclinical at-risk patients and those with early disease requires combining the existing “A-T-N” paradigms with the new preclinical testing measures on a timeline correlating disease stage and test conversion to positive. See Table 2 (below)

## 14. Current Treatment Trials and the Future

The possibility of treatment trials addressing the BBB leak in AD are under consideration. If subjects in the preclinical or very early symptomatic phase of AD are considered for treatment trial, then a strategy for developing a testable hypothesis must be in place. One method would be to screen (using the above MRI techniques) asymptomatic at-risk population for developing BBB leak and impaired glymphatic flow such as individuals with AODM of 10 years duration, hypertensives, strong family history of dementia etc. Those with positive studies could be randomized into treatment and non-treatment arms with baseline and follow-up “A-T-N” (pathologic) testing and NP (clinical) testing. Marker conversion from negative to positive would indicate disease progression. The clinical correlation could be followed by repeat NP and other functional measures. Reversion to normal physiology and absence of pathologic and clinical marker evolution could be followed and compared. This would shorten the trial duration substantially reducing costs and allowing more rapid community access to effective treatments.

Borrowing potential treatments in stroke trials and other disease processes may open new arenas for early AD treatment [21]. Since these BBB changes develop well before the proteinopathy, identifying subjects at highest risk to enter into investigative trials in the pre-clinical or early clinical stage of illness is paramount to assessing treatment efficacy. Although the pericytes appear to be the first casualty in the process, their dysfunction may still be triggered remotely given their complex signaling interrelationship and interdependence with endothelial cells and astrocytes [29]. The cascade of compounding destructive processes such as leaking iron, and other toxins could perhaps be shut off with a “stop leak” treatment. An example of possible mechanisms to potentially stop the BBB leak is employing activated protein C that has completed phase 2 trials (dose related and tolerability) in stroke treatment. Whether this strategy could be employed successfully in low leak situations may be a worthwhile consideration [21].

CypA inhibitor of the CypA-MMP-9 Pathway-induced BBB tight junction damage is completing phase 3 trial in treatment of hepatitis C, and could be considered for clinical trial, especially since there were minimal identified risks in the hepatitis trial [21].

Methods of enhancing astrocyte redeployment of aquaporin-4 water channels to the astrocytic end feet to improve paravascular clearance thereby “unclogging the drain” are potential targets as well [91]. Reducing flow-obstructing precipitated toxic proteins from the interstitium may improve glymphatic clearance but remains to be seen.

Whether early BBB leak directly causes the later development of toxic protein misfolding by epigenetic signaling pathways via pericytes, endothelial cells, or astrocytes remains unknown. Alternatively, the leak may allow passive influx of prion-like templates from a distant source such as the gut. Further, investigating gut flora in AD is under investigation NCT04100889. Of interest in this regard is whether the enteric system could be a source for blood borne HP Tau prion-like template. Should this be the case then repairing the BBB leak early in the preclinical or early clinical disease state becomes all the more urgent.

One effort to reduce blood born amyloid β-protein reflux using RAGE blocker was initiated but terminated in PHASE 3 trials due to lack of efficacy.

In a recent review the penetration of large biologic molecules through the BBB remains a major obstacle for delivering efficacious drugs [3]. Pioneering work is ongoing, adapting nanoparticle technology for delivery of contrast agents, or therapeutic ligands to specific layers of the neuropil or neurovascular compartment [92].

## 15. Conclusions

It is clear that successful treatment of AD will require very early intervention before tau-related neurodegeneration arises. Identification of BBB leak and possibly concomitant glymphatic flow dysfunction in preclinical AD (preceding accumulation of either Aβ or Hp-tau), requires use of reliable non-invasive tools to monitor results of future treatment strategies aimed at “stopping the leak” and “clearing the drain.” The MRI tools discussed above could serve this role. Creating a coherent testable hypothesis proving treatment efficacy will need to incorporate both pathologic diagnostic criterion with new and established associated testing strategies, along with clinical outcome measures. By doing so, the treatment trial duration could be shortened, and effective treatment strategies could be approved more rapidly for clinical use.

## Figures and Tables

**Figure 1 biomedicines-08-00228-f001:**
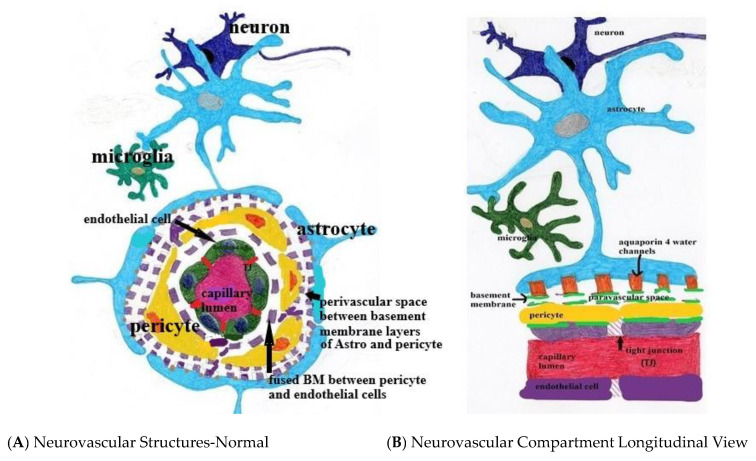
(**A**,**B**) Normal neurovascular anatomy. Note the tight junctions (TJ) between endothelial cells and the fused two-layer basement membrane (BM) separating them from surrounding pericytes. The BM separates into two layers between the pericytes and astrocyte foot processes forming the perivascular space. Aquaporin 4 water channels expressed in the astrocyte foot process allow bidirectional water flow within the astrocytes and the paravascular space. Egress of paravascular fluid along paravenous channels carries with it actively transported waste from the interstitium ultimately into the meningeal lymphatics.

**Figure 2 biomedicines-08-00228-f002:**
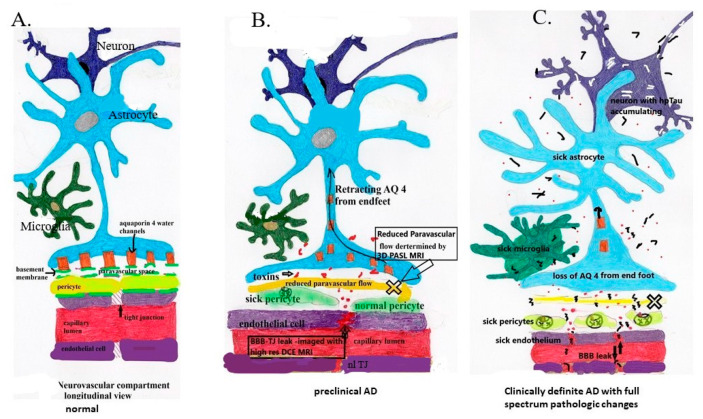
Note the pathologic sequence leading to sporadic mild and advanced AD. (**A**) Demonstrates normal anatomy and physiology. (**B**) demonstrates the development of low-grade BBB leak and retraction of AQ 4 with reduction in paravascular flow prior to accumulation of Aβ and Hp tau. (**C**) demonstrates chronic persistent vascular leak and paravascular outflow obstruction with now accumulation of first Aβ1 – 42 oligomers and subsequent intraneuronal accumulation of HpTau with potential prion-like transsynaptic dispersal. In flowing toxins including perivascular and intravascular Aβ. 
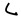
 Aβ, 
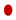
 infiltrating vascular toxins, 
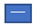
 Hp tau.

**Table 1 biomedicines-08-00228-t001:** Current pathologic criterion for AD diagnosis.

A = Aggregated Amyloid Aβ	T = Aggregated Tau	N = Neurodegeneration
CSF Aβ_42_ or Aβ_42_/ Aβ_40 ratio_	CSF phosphorylated tau	Anatomic MRI -atrophy
Amyloid PET	Tau Pet	FDG PET

A-T-N criterion and tests currently accepted under each category for research related AD pathologic diagnosis (independent of clinical diagnosis).

**Table 2 biomedicines-08-00228-t002:** Pathologic test sensitivity at respective stage of AD.

Test	Preclinical Stage	Early Clinical (MCI)	Definite AD (Late)
DCE	+	+	+
ASL flow	?	+	+
CSF Aβ_42_ or Aβ_42_/ Aβ_40 ratio_	±	+	+
CSF phosphorylated tau	-	+/-±	+
Amyloid PET	±	+	+
Tau Pet	-	+/-±	+
Anatomic MRI -atrophy	-	-	+
FDG PET	-	±	+
CSF total tau	-	-	+

Note the progression of disease process from preclinical to clinically definite AD and the presence of associated biologic markers. + = positive test, ± = negative test, ? = unknown at this time.

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
