# Peer review of "Novel MRI Techniques Identifying Vascular Leak and Paravascular Flow Reduction in Early Alzheimer Disease"

_biomedicines, 2020, doi:10.3390/biomedicines8070228_

Round 1

Reviewer 1 Report

The topic of this review is an important question in the field. However, more background descriptions about Alzheimer's disease, pathology and etiology may be helpful. 

Author Response

Reviewer 1 Thank you for your time and attention reviewing my manuscript.  I have greatly expanded the discussion of Alzheimer disease in the introduction and elsewhere.  Likewise I have detailed the clinical and pathologic diagnostic approaches to diagnosis and the rationale for the differences.  Nine additional references were added to that point.

Reviewer 2 Report

  • Is the primary amyloid receptor mentioned in the text LPR1 or LRP1? Kindly check.
  • Author can present the existing diagnostic methods of Alzheimer’s disease and the reason for their partial success in a tabular form for better understanding of the advantages of suggested MRI techniques over them. Alternatively, he can also provide a comparative analysis of novel MRI techniques mentioned with the existing disease diagnosis and treatment methods.
  • The author is requested to consider revising lines 334-348 under the heading 3D PASL MRI since they have been directly lifted from his previously published paper.
  • The conclusion portion in the text is too concise. Maybe the author can add some more points highlighting the overall significance of MRI techniques in context of Alzheimer’s disease.
  • Adding a subheading explaining preclinical and clinical significance of the mentioned MRI techniques would provide more substance to the topic.
  • The headings for figure 1 and 2 are not required. The font for legend of figure 1 can be increased for better readability.
  • The figures 2A-C are not descriptive enough and a bit difficult to comprehend. Could the authors provide cleaner images/figures?
  • The entire manuscript needs a thorough grammatical proofread.

Author Response

#1 LRP1 is correct and has been changed

#2 I have greatly expanded the current criterion and methodology of pathologic diagnosis per NIA-AA guidelines and contrasted it with current clinical criterion.  The pre-clinical diagnostic imaging studies identifying the BBB leak and reduced glymphatic clearance are also integrated. New tables and sections have been added to illustrate pathologic criterion for treatment trial  design and implementation.

#3 Lines 248-348 have been modified and simplified regarding the specifics of my MRI methodology to include the bare essentials with reference to my previous study for details.

#4 The conclusion has been expanded to include additional points to include rationale for adding pre-clinical criterion to future treatment protocols and need to treat AD early.                                                                                             #5 Subheading  re: Pre-clinical diagnostic testing rationale has been added.                                                                                                                       #6 Figure 1 has been modified and its caption per request.                           #7 Figure 2a-c and caption have been  modified as requested                        #8 I have had the manuscript thoroughly proofread.                                           I wanted to thank you for the excellent review of my manuscript.  You definitely gave me clearer insight into developing new methodology to diagnose, enroll subjects with pre-clinical AD, and follow the results of new treatment strategies.                                                                                                    I found this text box particularly difficult to use as it would not allow cut and paste as usual despite using the prompts...can it be fixed?